# The Immune Checkpoint PD-1 in Natural Killer Cells: Expression, Function and Targeting in Tumour Immunotherapy

**DOI:** 10.3390/cancers12113285

**Published:** 2020-11-06

**Authors:** Linda Quatrini, Francesca Romana Mariotti, Enrico Munari, Nicola Tumino, Paola Vacca, Lorenzo Moretta

**Affiliations:** 1Department of Immunology, IRCCS Bambino Gesù Children’s Hospital, 00146 Rome, Italy; fromana.mariotti@opbg.net (F.R.M.); nicola.tumino@opbg.net (N.T.); paola.vacca@opbg.net (P.V.); lorenzo.moretta@opbg.net (L.M.); 2Department of Molecular and Translational Medicine, University of Brescia, 25121 Brescia, Italy; enrico.munari@unibs.it

**Keywords:** PD-1, PD-L1, NK cells, inhibitory checkpoints, tumour microenvironment, immunotherapy, soluble PD-1, glucocorticoids

## Abstract

**Simple Summary:**

Natural killer cells are innate cytotoxic lymphocytes that play a key role in the anti-tumor immune response. In the tumor microenvironment, however, the effector functions of these cells are often impaired by the induction of inhibitory surface molecules, including PD-1. In the present review, we provide further insight into the expression and function of the immune checkpoint PD-1 in natural killer cells, together with the limitations and perspectives of immunotherapies aimed at blocking the interaction of this inhibitory receptor with its ligands.

**Abstract:**

In the last years, immunotherapy with antibodies against programmed cell death protein 1 (PD-1) and programmed death-ligand 1 (PD-L1) has shown remarkable efficacy in the treatment of different types of tumours, representing a true revolution in oncology. While its efficacy has initially been attributed only to unleashing T cell responses, responsivity to PD-1/PD-L1 blockade was observed in some tumours with low Human Leukocyte Antigen (HLA) I expression and increasing evidence has revealed PD-1 surface expression and inhibitory function also in natural killer (NK) cells. Thus, the contribution of anti-PD-1/PD-L1 therapy to the recovery of NK cell anti-tumour response has recently been appreciated. Here, we summarize the studies investigating PD-1 expression and function in NK cells, together with the limitations and perspectives of immunotherapies. A better understanding of checkpoint biology is needed to design next-generation therapeutic strategies and to improve the clinical protocols of current therapies.

## 1. Introduction

Natural killer (NK) cells are innate lymphocytes that play an important role in anti-tumour immune responses. NK cells are capable of recognizing and killing tumour cells thanks to the expression of activating receptors binding ligands induced or upregulated by cell stress, particularly in tumours [1]. At the same time, the expression of inhibitory receptors recognizing Human Leukocyte Antigen (HLA) class I molecules allows NK cells to spare healthy cells and selectively kill those that underwent malignant transformation and may have lost HLA class I expression [2]. The central role of NK cells in tumour clearance and maintenance of long-term remissions is highlighted by studies showing that NK cell alterations in the tumor microenvironment (TME) are associated with adverse clinical outcomes [3]. NK cell dysfunction is associated with significant phenotypic alterations, including a decreased expression of the natural cytotoxicity receptors (NCRs) in solid tumors and hematological malignancies, and the downregulation of the activating receptors DNAX Accessory Molecule-1 (DNAM-1) and Natural Killer Group 2D (NKG2D) [4,5,6,7]. Moreover, impaired NK cell function in the TME has been associated with the increased expression of inhibitory receptors. Among these, a number of receptors have been identified and studied, which do not bind HLA class I molecules, but instead ligands expressed by tumour cells, favouring a mechanism of evasion from immune recognition [8]. These inhibitory receptors include Programmed cell death protein 1 (PD-1), Lymphocyte Activation Gene 3 (LAG3), T cell immunoglobulin and mucin-domain containing-3 (TIM3) and T cell immunoreceptor with Ig and ITIM domains (TIGIT). They are expressed by both NK and cytotoxic T lymphocytes and, because they act as gatekeepers of immune responses, they have been named “immune checkpoints”. NKG2A (an inhibitory receptor binding HLA-E) has also been recognized as an immune checkpoint [9]. It was shown that its expression is increased in NK cells and induced in T cells in the TME, and that it is co-expressed with PD-1 in both cell types [10].

As surface molecules, immune checkpoints’ activity can be inhibited by blocking antibodies that prevent ligand-receptor engagement. The most successful immune checkpoint blockade therapy to date is anti-PD-1/PD-L1 therapy, currently approved to treat a wide variety of cancer types [8]. While it was initially thought that blocking PD-1/PD-L1 axis would only unleash T cell response, it is now becoming clear that NK cell responses may also be potentiated through this strategy. Therefore, it is important to better investigate the expression and function of this receptor on NK cells to fully exploit their anti-tumour response.

In this review, we will summarize the data on the biology of PD-1 protein, its expression by NK cells and the strategies developed to block PD-1/PD-L1 in immunotherapy, including their limitations and future perspectives.

## 2. The PD-1 Immune Checkpoint

PD-1 was originally discovered as an apoptosis-associated gene during T cell thymic selection [11]. Further studies revealed that the physiologic role of PD-1 is not related to cell death, but it is instead involved in regulation of immune responses, as demonstrated by the development of lupus-like autoimmune diseases in PD-1-deficient mice [12,13]. PD-1 belongs to the CD28/Cytotoxic T-lymphocyte-associated protein 4 (CTLA-4) subfamily of the Immunoglobulin (Ig) superfamily [14] and it is a 50–55 kDa type I transmembrane glycoprotein [15]. PD-1 specifically binds the type I transmembrane proteins PD-L1 (B7-H1; CD274) [16,17] and PD-L2 (B7-DC; CD273) [18,19], both of which are B7 family members [20,21]. PD-L1 is constitutively expressed in a wide range of cells, including hematopoietic and non-hematopoietic cells. In contrast, PD-L2 expression is restricted to professional antigen-presenting cells (APCs; monocytes, macrophages, and dendritic cells (DCs)) and to a subset of B cells. Inflammatory cytokines such as interferons (IFNs; α, β, and γ) are potent regulators of both PD-L1 and PD-L2 expression. [22,23].

From the structural point of view, PD-1 consists of a single Ig variable-type (IgV)-like domain in the N-terminal region, a stalk of approximately 20 amino acids required to separate the IgV region from the plasma membrane, a transmembrane domain and an intracellular cytoplasmic tail acting as docking site for protein recruitment and signalling [24]. Indeed, two important structural motifs have been found in the intracellular PD-1 region: the immunoreceptor tyrosine-based inhibition motif (ITIM) and the immunoreceptor tyrosine-based switch motif (ITSM). Although ITIM is usually present in inhibitory receptors, mutational studies have shown that the ITSM domain might play a pivotal role in promoting PD-1 inhibitory function [25]. Upon PD-1/PD-Ls axis activation, phosphorylation of the ITSM (Y248 residue) domain is required for Src homology region 2 domain-containing phosphatase-2 (SHP-2) recruitment; interaction of PD-1 with SHP-2 is essential for downstream signalling pathway modulation. Phosphorylation of the Y248 residue within the ITSM domain has been detected on T cells from mouse tumour-draining lymph nodes and also in biopsies of glioblastoma patients [26]. Thus, Y248 phosphorylation might represent a biological biomarker able to identify immune cells that are actively immunosuppressed due to PD-1 pathway activation. Despite the fact that SHP-2 plays a pivotal role in regulating PD-1 inhibitory pathway, the precise mechanism of interaction is yet to be clarified. Two different PD-1/SHP-2 interaction models have recently been proposed. According to the two-step binding model, and in contrast to previous observations, phosphorylation at both the ITSM and ITIM domains is required for SHP-2 recruitment and activation of the downstream signalling cascade [27]. In contrast, the dimeric model proposes that SHP-2, through its N- and C-terminal domains, bridges to phosphorylated residues on two PD-1 molecules forming a PD-1:PD-1 dimer, which can induce a more robust SHP-2 activation [28]. Triggering of PD-1 promotes the insurgence of an exhaustion phenotype in both innate (NK and innate lymphoid cells (ILCs)) and adaptive immune cells with a consequent blockade of proliferation and cytotoxic activities. The molecular pathways involved in immune cell inhibition have been identified in T cells. Indeed, it has been shown that triggering of PD-1 and SHP-2 recruitment activates a signalling cascade that modulates the PI3K-Akt and Ras-MEK-ERK pathways dampening proliferation, differentiation and survival (Figure 1) [29]. However, PD-1 engagement was shown to downregulate STAT5 phosphorylation in ILC2s, inhibiting their proliferation and function [30]. In addition, PD-1 signalling activation modulates immune cells metabolism, which has an inhibitory effect on immune cells’ differentiation and proliferation. It is well known that, upon activation, immune cells reprogram cell metabolism towards aerobic glycolysis. However, engagement of the PD-1 pathway suppresses oxygen consumption and impairs glycolysis while favoring the utilization of fatty acid in β-oxidation (FAO) [29]. Though PD-1-dependent metabolic reprogramming has been demonstrated in T cells, little is known of its role in modulating NK cells’ metabolism in the tumour context [31,32,33]. Therefore, considering the extensive inhibitory responses driven by PD-1 triggering, the PD-1/PD-Ls axis represents a molecular mechanism widely exploited by tumour cells to escape recognition by the immune system.

## 3. Evidence for PD-1 and PD-L1 Expression on NK Cells

PD-1 function was initially best characterized in T cells, where it plays an immunoregulatory role responsible for limiting excessive activation and the prevention of immune-mediated damage. PD-1 expression on T cells is induced by activation-driven T cell receptor (TCR) signalling and further up-regulated by cytokines [22]. On the other hand, chronic TCR stimulation and PD-1 expression lead to T cell dysfunction. Importantly, tumour-infiltrating T cells express high levels of PD-1 due to prolonged exposure to tumour antigens and the immunosuppressive environment, and exhibit functional and phenotypic properties similar to the exhausted T cells present in chronic infections. These include defects in effector cytokine production and up-regulated expression of inhibitory receptors [34,35,36]. Therefore, PD-L1 expression by tumour cells represents a mechanism of immune evasion to avoid killing by PD-1^+^ cytotoxic lymphocytes. The first evidence of a role for PD-1 in tumour immune evasion came from a study showing that transgenic PD-L1 expression by a tumour cell line rendered it less susceptible to the specific TCR-mediated lysis by cytotoxic T cells in vitro, and markedly enhanced its tumorigenesis and invasiveness in vivo in the syngeneic hosts as compared to the parental tumour cells [37]. PD-L1 is indeed expressed by many types of cancer cells and also by myeloid cells present in TME, and its expression is up-regulated by various inflammatory stimuli in the TME [37,38,39]. Studies in vivo suggested that blocking PD-1/PD-L interaction could provide a promising strategy for specific tumour immunotherapy [37,40]. Indeed, encouraging results from preclinical studies led to the development of several clinical-grade blocking antibodies against PD-1 or PD-L1 (see Section 5.1).

The currently recognized mechanism underlying the PD-1/PD-L1 axis blockade in tumour sites is that the interaction of PD-L1 on tumour cells with PD-1 on tumour-infiltrating T lymphocytes delivers negative signals and inhibits antitumor T cell response, facilitating tumorigenesis. However, many cancer types exhibit a high incidence of HLA loss and/or low neoantigen burden, which render tumour cells refractory to recognition by CD8^+^ T cells. High levels of PD-L1 expression have been detected in tumours with low HLA I expression, which, in some instances, are responsive to PD-1/PD-L1 blockade [41,42,43]. For example, in Hodgkin’s lymphoma, decreased/absent expression of HLA-I expression on malignant cells is detected in the majority of patients [44]. Nevertheless, in a study analyzing the effect of PD-1/PD-L1 blockade in this type of tumor, an objective response was reported in 87% of patients enrolled, including 17% with a complete response and 70% with a partial response [45]. These results suggest that T-cell-independent cytotoxic anti-tumour immune responses exist, that are inhibited by PD-1 and rescued by PD-1 blockade in Hodgkin’s Lymphoma. A similar deduction is probably applicable for some other types of human tumors. This hypothesis is consistent with the more recent finding of the expression of PD-1 on NK cells, displaying cytotoxic activity against cancer cells. NK cells lack the expression of antigen-specific receptors, but express an array of germline-encoded receptors that allow the recognition of cells expressing ligands induced by cell stress and neoplastic transformation (induced-self) and cells that have lost the expression of major histocompatibility complex (MHC) molecules (missing-self) [46]. Different from T cells, NK cells present in the peripheral blood (PB) of healthy individuals do not express PD-1 on their surface, with the exception of a fraction of cytomegalovirus (CMV)^+^, otherwise normal, subjects [47]. However, evidence that PD-1 can be induced on human NK cells has recently emerged in several cancers (Table 1). PD-1 expression on NK cells was detected in the PB of Multiple Myeloma [48] and renal cell carcinoma patients [49]. In renal cell carcinoma, PD-1 expression seemed to be restricted to the CD56^dim^ subset and correlated with disease stage and an activated NK cells phenotype [49]. In Kaposi Sarcoma, a sub-population of activated, mature CD56^dim^ NK cells expressing PD-1 was identified, with otherwise normal expression of NK surface receptors [32]. PD-1 expression was also detected on NK cells in the peritoneal fluid of ovarian carcinoma patients [47] and high-grade peritoneal carcinomatosis [50], and on circulating CD56^bright^ NK cells in Hodgkin Lymphoma patients [51]. Moreover, it was reported that PD-1 expression on NK cells significantly correlates with poor prognosis in digestive cancers [52]. In addition, a subset of PD-1^+^ tumour-infiltrating NK cells was identified and characterized in non-small cell lung cancer (NSCLC): PD-1^+^ NK cells co-expressed more inhibitory receptors as compared to the PD-1^−^ subset, and increasing levels of PD-1 expression correlated with intratumoural NK cell dysfunction [53]. It was also shown that both NK cells and ILC3 expressed functional PD-1^+^ in the pleural effusions of patients with primary or metastatic lung cancer [54]. Notably, PD-1 expression was reported, in addition to cytotoxic ILCs, on helper ILCs and in contexts other than cancer [55,56]: on a committed ILC progenitor in mouse bone marrow [57] and on ILC3s in human decidua [58]. In particular, it was shown that ILC3 expresses both PD-1 and TIM-3 during the first trimester of pregnancy and that these receptors could regulate production of cytokines, suggesting that PD-1/PD-L1 interaction may also regulate tolerance at the feto-maternal interface [58]. PD-1 expression on NK cells was shown also in many viral infections: in murine hepatitis virus strain-3 (MHV-3) [59], chronic HIV-1 [60] and murine CMV (MCMV) infection [61].

The first mechanistic evidence for PD-1 as an important checkpoint for NK cell activation was found in MHC-deficient tumours: PD-1 expression was detected at a very early timepoint (48 h) after tumour inoculation on a subset of intratumoural NK cells [62]. In this study, a significant contribution of NK cells to immunotherapy mediated by PD-1/PD-L1 blockade was convincingly demonstrated in transplantable, spontaneous or genetically induced mouse tumour models [62].

In a recent report, Judge and colleagues [63] failed to reproduce the results of other groups as they could not show PD-1 expression on NK cells in viral and tumour models [63]. The authors checked PD-1 expression on murine, canine and human NK cells in parallel with other activation and exhaustion markers using multiple techniques, including flow cytometry, quantitative real-time PCR and RNA sequencing. They repeatedly demonstrated that NK cells in many solid tumour murine models and in MCMV infection, and human NK cells activated in vitro and isolated from cancer patients’ samples, all display minimal PD-1 expression [63]. Therefore, they concluded that, because of the minimal PD-1 expression on NK cells, their contribution to immunotherapy may not be significant and PD-1/PD-L1 pathway does not contribute to NK cell immunoregulation.

The contrasting results between the study by Judge and the previous works indicated that a rigorous methodology is needed to avoid inconsistencies in how investigators measure PD-1 expression. On the other hand, in addition to the detection of PD-1 expression on NK cells, many of the previous studies included functional experiments with anti PD-1 agents to verify the effect on NK cell activity, which indirectly confirmed their data. Moreover, the timing of PD-1 expression appears to be crucial for its detection on NK cells, as it is very different as compared to T cells. Evidence demonstrates that PD-1 expression is not associated with NK cell exhaustion but rather with acute activation. This may explain why more robust PD-1 expression can be observed on NK cells upon stimulation, while T cells maintain PD-1 on their surface as a hallmark of early antigen-specific activation and later chronic stimulation. PD-1 expression appears to be rapid and transient on NK cells and occurs at the time needed to prevent excessive NK cell activation. For example, upon MCMV infection PD-1 is expressed only on NK cells in the spleen and at an earlier timepoint compared to the one analysed by Judie and colleagues [61]. Finally, a critical factor driving PD-1 expression on NK cells is represented by glucocorticoids (GCs) (see Section 4), which the authors did not test in their in vitro experiments.

Many studies reported that PD-L1 can also be induced on either transformed or healthy NK cells. PD-L1 expression on malignant NK cells was detected by immunohistochemistry in two of eight cases of EBV^+^ [64] and two of three cases of EBV^−^ [65] aggressive natural killer-cell leukemia (ANKL), and copy number gains of PD-L1- and PD-L2-encoding genes were found in NK/T cell lymphoma (NKTCL) patients [66]. Moreover, it was reported that “non malignant” NK cells express PD-L1 in mouse models of viral infection [67,68] and tumour [69,70]. Notably, it was shown that myeloid leukemia cell lines and acute myeloid leukemia blasts from patients can induce PD-L1 on human NK cells via AKT signaling [71]. It was shown that engaging PD-L1 with anti-PD-L1 monoclonal antibodies (mAbs) resulted in enhanced NK cell anti-tumor activity via a p38 pathway [71]. The discovery of a PD-1-independent mechanism of anti-PD-L1 mAb antitumor efficacy via the activation of PD-L1^+^ NK cells provides a potential explanation to why some patients lacking PD-L1 expression on tumor cells still respond to anti-PD-L1 mAb therapy.

It is recognized that the PD-1 pathway is an important determinant of the outcome of the T cell response, and studies in different contexts (tumours and infections) show that the role of this pathway in T cells is to regulate the balance between effective host defence and immunopathology. All the data collected over the recent years on PD-1 expression on NK cells suggest that the PD-1 pathway may play the same role in these innate lymphocytes, and the kinetics and duration of PD-1 expression parallel the kinetics of NK cell activation and response. Indeed, different from T cells, the complexity of the microenvironment provides dynamic changes in PD-1 expression on NK cells, which are species-specific and clinical condition (tumour- or infection)-specific. Because different splicing variants and a soluble form of PD-1 exist (see Section 5.2), PD-1 induction on NK cells may also vary in different individuals.

Because of the different contribution of adaptive and innate cytotoxic lymphocytes in anti-tumour response, it is crucial to further dissect the molecular mechanisms, the timing and the clinical conditions driving PD-1 expression on NK cells in depth, to better understand how to unleash NK cells’ anti-tumour function by blocking the PD-1/PD-L1 activity.

## 4. Mechanisms that Control PD-1 Expression on NK Cells: An Indispensable Role for Glucocorticoids

The interaction of PD-1 with its ligands in the tumour context inhibits both innate and adaptive anti-tumour immune responses, promoting tumour growth and metastatic spreading. Thus, it is not surprising that PD-1 expression is tightly regulated. Indeed, several transcriptional factors are involved in the induction and repression of the *PDCD-1* gene, ensuring that this inhibitory checkpoint is expressed in a finite window of time [29]. While it is clear that PD-1 expression on T cells is dependent on TCR engagement, the mechanisms regulating the de novo PD-1 induction on NK cells has been investigated only recently. It has been shown that resting human NK cells express PD-1 transcript and intracellular protein localized in the Golgi, but express only minimal levels of surface receptors [73]. The presence of this intracellular pool would suggest that PD-1 can be rapidly expressed on the cell surface membrane and inhibits NK cell activation in response to given stimuli. To date, the steroid hormones glucocorticoids (GCs) have been identified as an indispensable stimulus required for PD-1 surface expression on both murine and human NK cells [61,72]. These hormones are secreted by the adrenal gland into circulation in response to stimulation of the hypothalamus–pituitary–adrenal (HPA) axis by stress, and inflammatory cytokines released systemically [74]. The general role of this axis is to suppress excessive inflammation in a negative feedback loop, and the induction of immune checkpoints on lymphocytes has been identified as an additional immune suppressive mechanism [74,75]. In a mouse model of infection with MCMV, it was shown that at the peak of the HPA axis activation GC receptor (GR) induces PD-1 expression on spleen NK cells, thus inhibiting IFN-γ production in this organ. This GC-PD1-IFN-γ axis was shown to be indispensable for host protection from the deleterious effects of hyperinflammation induced by NK cell-mediated anti-viral response. Mechanistically, PD-1 expression on NK cells was shown at the transcript and protein level, and the dependence on GC was demonstrated by comparing in vivo NK cells expressing or not expressing the GR. Moreover, it was shown in vitro that GCs alone are not sufficient to induce PD-1 on spleen NK cells, but GR signaling is integrated to the signals transduced by IL-15 and IL-18, the most abundant cytokines present in the organ upon MCMV infection [61]. Given the importance of the PD-1 pathway in the context of cancer immunotherapy, it was then investigated whether GCs could also induce PD-1 in human NK cells. Interestingly, repeating the in vitro experiments previously done on murine spleen NK cells on human NK cells isolated from PB mononuclear cells revealed important differences between the two species. While PD-1 was induced after 48 h of stimulation on mouse NK cells, PD-1 induction on human NK cells required 6 days and was transient, dropping at day 10 [72]. Moreover, IL-15 and IL-18 stimulation, in combination with GCs, was not sufficient to induce PD-1 on human NK cells, but IL-12 was also required. Notably, the addition of this cytokine completely abolished GC-dependent PD-1 induction on mouse NK cells. Therefore, not only the kinetics of PD-1 induction by GCs are different between the two species, but also the combination of cytokines required. In addition, parallel analysis of PD-1 transcript and protein expression upon GC and cytokine stimulation showed that, in human NK cells, PD-1 is induced not only at the transcript level, but also at a post-transcriptional level by the activation of a transcriptional program leading to enhanced protein translation and translocation to plasma membrane [72]. It was also shown that the release of endogenous GCs was increased in the plasma of lung cancer patients in comparison to healthy donors, and this increase was associated with high concentrations of this hormone at the tumour site. However, GCs alone are not sufficient for PD-1 induction on NK cells, and the cytokines present in the TME (including IL-12, IL-15, and IL-18) are fundamental. Indeed, the proportion of PD-1^+^ NK cells was higher at the tumour site compared to the PB [54,72]. Similarly to what was shown for NK cells in murine cancer models [62], this study on human NK cells also confirmed that PD-1 expression is not associated to exhaustion but rather to cell activation: PD-1^+^ and PD-1^−^ NK cells could comparably respond to cytokine stimulation and triggering of activating receptors, while only direct PD-1 engagement had an inhibitory effect [72].

The effect of GCs on PD-1 expression was demonstrated also in T lymphocytes [76], where transactivation of PD-1 transcription by GR binding to the *PDCD1* promoter was reported [77]. Notably, unlike NK cells, GCs further increase TCR-dependent PD-1 induction, but are not required for it.

The identification of GCs as important factors in driving PD-1 expression on cytotoxic lymphocytes has crucial clinical implications. Indeed, corticosteroids are frequently given to oncologic patients to help tolerate the effects of chemotherapy and immunotherapy, and some studies show a possible role of GCs in anti-PD-1/PD-L1 resistance. For instance, it was shown that patients with NSCLC treated with prednisone at the time of anti-PD-1/PD-L1 immunotherapy (alone or in combination with CTLA4 blockade) have worse outcomes than control patients [78]. Similarly, in two independent cohorts of NSCLC patients treated with a single-agent PD-(L)1 inhibitor, baseline corticosteroid treatment was associated with decreased overall response rate, progression-free survival, and overall survival [79]. Therefore, it must be taken into account that GCs may not only contribute to suppress recognition of tumours expressing PD-Ls, but may even reduce the efficacy of immunotherapy targeting the PD-1/PD-L1 pathway.

## 5. Targeting the PD-1/PD-L1 Pathway in Tumour Immunotherapy: State of the Art, Limitations and Future Perspectives

### 5.1. Immunotherapy with Antibodies Against PD-1 and PD-L1 

Since the approval by the Food and Drug Administration (FDA) of anti PD-1 nivolumab and pembrolizumab in 2014 for the treatment of advanced melanoma [80,81], researchers and companies have put enormous effort into finding and developing novel and effective agents targeting the PD-1/PD-L1 axis. Thousands of trials have been then carried out, many of them showing positive results and leading to the approval of several antibodies targeting the PD-1/PD-L1 axis in different neoplastic settings, including anti PD-1 nivolumab (Opdivo), pembrolizumab (Keytruda), cemiplimab (Libtayo) and anti PD-L1 atezolizumab (Tecentriq), durvalumab (Imfinzi) and avelumab (Bavencio) (Table 2).

The number and types of advanced or progressed tumours that can be treated with PD-1/PD-L1 axis inhibitors is rapidly and constantly growing as more and more clinical trials highlight their usefulness in several settings. One of the most challenging issues in this field is the selection of patients who may benefit the most from anti PD-1/PD-L1 therapies, since a relevant percentage of patients do not respond to these treatments. For this reason, much effort has been taken to develop biomarkers with predictive potential. The most known and used biomarker to date is PD-L1 expression. Thus, its expression within the tumour, evaluated with different methods, is required in defined settings for the administration of the anti-PD-1 pembrolizumab in NSCLC, head and neck squamous cell carcinoma (HNSCC), urothelial carcinoma, oesophageal carcinoma, gastric carcinoma and cervical carcinoma and of the anti-PD-L1 atezolizumab in triple-negative breast carcinoma (TNBC) and urothelial carcinoma. However, besides the intrinsic and complex dynamics of the tumour immune contexture, many factors can eventually hamper the predictive potential of PD-L1 expression and have to be taken into account in order to define strategies to improve the value of these biomarkers. Such variables include PD-L1 expression heterogeneity, differences in terms of PD-L1 expression between types of samples analysed (cytology vs. biopsies vs. surgical specimen), inter-clones and interobserver variabilities [82,83,84,85]. Moreover, it has been demonstrated that tumours with mismatch repair (MMR) deficiency respond better to PD-1/PD-L1 inhibitors, as they harbour genomic instability with high mutation rates and therefore are characterized by abundant neoantigens, which render them more immunogenic [86]. In fact, the presence of MMR deficiency has become a selection criteria for treatment with both pembrolizumab and nivolumab [87], and pembrolizumab has been approved as a second-line therapy, irrespective of the primary tumour location [88,89]. Following this line of reasoning, a high tumour mutational burden (TMB), given the presence of a high quantity of neoantigens, could be associated with response to immunotherapy [90]. The cancer immune contexture appears to be extremely important in order to understand the interplay between different types of effector immune cells and their relationship with tumour cells and their immune escape mechanisms [91]. The qualitative and quantitative evaluation of tumour-infiltrating lymphocytes, therefore, holds great promise in the identification of patients whose tumours will more likely respond to immunotherapy. Different studies evaluated the impact of CD8^+^ cell densities in NSCLC and response to anti PD-1 [92,93]. Given the growing evidence suggesting an impact of PD-1 expression on NK cells in the outcome of immunotherapy, it is foreseeable that the precise definition of the tumour immune contexture including these innate lymphocytes will become even more important in the future. The efficacy of PD-1/PD-L1 axis blockade in HLA I^−^ tumours [45], together with the finding that triggering PD-L1 on NK cells with anti-PD-L1 mAbs enhances anti-tumor response [71], suggests a relevant contribution of NK cells in the positive outcome of therapies targeting either the receptor or the ligand. Moreover, given the similar profiles of surface inhibitory receptors expression by T and NK cells, a promising strategy consists in the utilization of combined immunotherapies targeting multiple checkpoints besides the PD-1/PD-L1 axis, including NKG2A, TIM-3, LAG-3 and TIGIT [8]. It is expected that the number of therapies involving, but not limited to, the inhibition of the PD-1/PD-L1 axis will continue to rise, together with their clinical indications. It is therefore crucial to continue to further elucidate the immunologic mechanisms on the basis of tumour immune escape in order to refine and optimize strategies to expand the immune targets as well as overcome mechanisms of resistance with the aim of providing effective treatments for most cancer patients.

### 5.2. Soluble PD-1 as a Novel Target of Immunotherapy and Prognostic Factor

Several immune checkpoint receptors can exist not only as membrane-bound forms but also as soluble forms, which can derive from the translation of alternative splicing isoforms or from proteolytic cleavage of the membrane-bound form [24]. Five different splicing variants of PD-1 have been identified, and the one derived from exon3 skipping (PD-1 Δex3), which lacks the sequence for membrane localization, encodes for the soluble form of PD-1 (sPD-1) and was initially detected in activated PB mononuclear cells [73,138]. Recently, our group detected PD-1 Δex3 mRNA in both resting and activated NK cells, as well as in NK cells isolated from pleural effusion of primary and metastatic tumours, indicating that NK cells are potentially able to release soluble PD-1 [73]. However, the stimuli regulating sPD-1 release in NK cells, or other immune cells, have not yet been identified.

Recently, growing evidence indicates that this soluble form may exert important immune-regulating functions and can be considered as a novel target for the development of new immunotherapeutic strategies [139]. Indeed, several studies performed on murine models shed light on the functions of sPD-1 in regulating CD8^+^ T-cell-mediated anti-tumour immune response. Similarly, enhanced cancer cells lysis and prolonged overall survival (OS) was observed in tumour-bearing mice upon delivery of sPD-1 into tumour sites [140]. Improvement in anti-tumour immunity was due to sPD-1 blockade of the PD-1 pathway, indicating that sPD-1 competes in vivo with PD-1 for binding to its ligands and represents a potent regulator able to unleash both the anti-tumour immune response [141,142].

mAbs are widely used to treat different tumour types and have been shown to improve NK-dependent anti-tumour immune activity [48,62,143]. Recently, it has been proposed that sPD-1 might instead be more effective than mAbs in restoring the immune response. This is due to the ability of sPD-1 to suppress all three combinations (PD-L1/PD-1; PD-L2/PD-1; PD-L1:B7-1) known to inhibit the immune response and also to the fact that sustained serum sPD-1 levels can still exert strong therapeutic effects while reducing the side effects that often occur with the use of anti-PD-1 mAbs [144]. In addition, considering that not all patients benefit from mAbs therapies, the identification of prognostic markers is becoming strictly required [145,146].

In this context, sPD-1 is considered a potential biomarker for evaluating prognosis and severity of cancers as well as response to treatment [146]. Indeed, it has been demonstrated in NSCLC and pancreatic cancer patients that sPD-1 serum level is associated with OS and progression-free survival (PFS), and it can be considered a predictive survival marker [147,148,149,150]. Most importantly, different studies reported that analysis of circulating sPD-1 could predict the efficacy of anti-PD-1 mAb treatment, allowing discrimination between responders and non-responders [146,151]. Due to sPD-1 ability to bind mPD-Ls, patients expressing high levels of sPD-1 might benefit less from anti-PD-1 therapy. In addition, sPD-1 could also interact with anti-PD-1 mAbs, limiting their bioavailability on tumour site; on the contrary, a lower affinity of sPD-1 toward mAbs might result in a better immune response [146].

A direct involvement of sPD-1 in unleashing PD-1 blockade on NK cells has not yet been demonstrated. However, considering that tumor-infiltrating NK cells express mPD-1 as well as the PD-1 Δex3 isoform, they might represent an ideal target for sPD-1 activity and investigating this interaction might allow for improvement in anti-PD-1 immunotherapies. It has recently been reported that a novel CAR-T cell construct, able to secrete sPD-1, acquired increased anti-tumour activity and cytotoxicity against CD19^+^PD-L1^+^-tumour cells due to the presence of sPD-1 [152]. These data indicate that sPD-1 can be used in combination with different immunotherapeutic approaches. CAR-NK cells, due to their potent cytolytic activity, represent a promising and powerful tool, and it will be interesting to evaluate if their combination with sPD-1 might improve their anti-tumour activity.

## 6. Conclusions

In the immune system responses to tumours, innate and adaptive immunity play complementary roles. Therefore, a therapeutic strategy able to simultaneously target the two branches of immunity has the potential to be greatly advantageous. This is the case for PD-1/PD-L1 axis blockade, which may restore the anti-tumour cytotoxic activity of both T and NK cells. This is particularly relevant because tumours often down-regulate HLA I expression as a mechanism of evasion from T cell recognition, but become more efficiently recognized and killed by NK cells. Blocking PD-1/PD-L1 pathway through mAbs or sPD-1 molecules unleashes T and NK cell response in different ways: while it reverts the T-cell-exhausted phenotype, PD1-expressing NK cells are not exhausted, and masking this receptor immediately restores their functional program.

All the evidence of PD-1 expression on NK cells and, in particular, the studies elucidating the underlying molecular mechanisms, allowed to shed light on the differences and similarities between the biology of PD-1 in NK and T cells. A deeper characterization of the immune context infiltrating the tumours (including NK cells and their expression of immune checkpoints) will help to better exploit the full potential of immunotherapy.

## Figures and Tables

**Figure 1 cancers-12-03285-f001:**
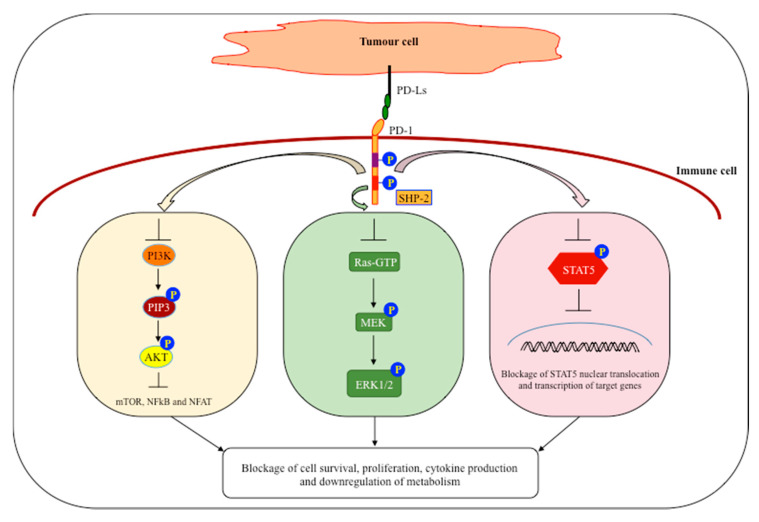
Major signalling pathways affected by the programmed cell death protein 1 (PD-1)/programmed death-ligands (PD-Ls) axis activation. In pathological conditions, the interaction between PD-1, expressed on immune cells, and its ligands upregulated by tumour cells, promotes inhibition of both the innate and the adaptive immune response. In particular, upon PD-1/PD-Ls complex formation, the tyrosine residues present in both the ITIM (purple rectangle) and ITSM (red rectangle) domains of PD-1 become phosphorylated and recruit SHP2, a key step for modulation of downstream molecular pathways. Indeed, PD-1 can block PI3K and Ras activation, which suppresses the signals delivered by the PIP3-AKT-mTOR and MEK-ERK MAP kinase pathways, respectively. In addition, inhibition of STAT5 phosphorylation impairs its nuclear translocation and blocks transcription of target genes. These inhibitory signals act together to modulate metabolic reprogramming and dampen cell proliferation, differentiation and survival as well as cytokine production.

**Table 1 cancers-12-03285-t001:** Evidence for PD-1 expression on human natural killer (NK) cells in tumours.

Tissue	Type of Tumour	Number of Patients	% Expression	Method	Reference
Peripheral blood	Multiple myeloma	5	nd	FC	Benson D.M., 2010 [48]
Peripheral blood	Renal cell carcinoma	90	nd	FC	MacFarlane A.W., 2014 [49]
Peripheral blood	Kaposi sarcoma	34	1–4	FC, IHC, qRT-PCR	Beldi-Ferchiou A., 2016 [32]
Peritoneal fluid/ascites	Ovarian carcinoma	30	10	FC	Pesce S., 2017 [47]
Peritoneal fluid	Low and high grade peritoneal carcinomatosis	6	5–10	FC	Pesce S., 2019 [50]
Peripheral blood and intratumoural	Hodgkin’s Lymphoma (HL) and diffuse large B-cell lymphoma (DLBCL)	66 HL 176 DLBCL	30 HL 10 DLBCL	FC	Vari F., 2018 [51]
Peripheral blood and intratumoural (HCC)	Digestive cancers (hepatocellular carcinoma; oesophageal squamous cell carcinoma; colorectal cancer)	4–18	5–10	FC, IHC	Liu Y., 2017 [52]
Intratumoural	Non-small cell lung cancer	11	5–20	FC	Trefny M.P., 2020 [53]
Pleural effusions	Primary and metastatic lung cancer	12	8–15	FC	Tumino N., 2019 [54] Quatrini L., 2020 [72]

The table summarizes the evidences reported for PD-1 expression on human NK cells in the context of cancer, including the tissue, the type of tumour and the reference of the studies. FC: flow cytometry; IHC: immunohistochemistry; qRT-PCR: quantitative real-time Polymerase Chain Reaction; HCC: hepatocellular carcinoma; nd: not determined (only mean fluorescence intensity is reported).

**Table 2 cancers-12-03285-t002:** Clinical indications for anti-PD-(L)1 immunotherapy in tumours.

Type of Tumour	Treatment	Therapeutic Indications [Ref]
Melanoma	nivolumab/pembrolizumab	I and II line [80,94,95,96]; adjuvant after complete resection [97,98]
	nivolumab combined	I line [99]
Esophageal cancer	pembrolizumab	II line if ≥10% tumor cells are PD-L1^+^ [100]
NSCLC	pembrolizumab	I line in metastatic tumour, EGFR and ALK wild type if ≥50% tumour cells are PD-L1^+^ [101,102] II line in tumors if ≥1% tumour PD-L1^+^ [103]
	atezolizumab combined	I line in tumours with non squamous histology, EGFR and ALK wild type [104]
	nivolumab/atezolizumab	II line [105,106]
	durvalumab	Stage III, non resectable tumours with no progression after chemoradiation [107]
Small cell lung carcinoma	atezolizumab combined	I line [108]
	nivolumab/pembrolizumab	II line [109,110]
Urothelial carcinoma	atezolizumab	I line for patients not eligible for cisplatin and tumours with ≥5% immune cells PD-L1^+^ or patients unfit for platinum-based therapy [111]
	pembrolizumab	I line for patients not eligible for cisplatin and tumours with PD-L1 CPS ≥10% or patients unfit for platinum-based therapy [112]
	atezolizumab/ pembrolizumab/nivolumab/durvalumab/ avelumab	II line [111,113,114,115,116]
Colorectal cancer	nivolumab (alone or combined)	II line in MMR-deficient cancer [117,118]
Gastric cancers	pembrolizumab	II line if PD-L1 CPS ≥1% [119]
HNSCC	pembrolizumab combined	I line in tumours with PD-L1 CPS ≥1% [120]
	pembrolizumab/nivolumab	II line [121,122]
Merkel cell carcinoma	pembrolizumab/avelumab	I line [123,124]
Cutaneous squamous cell carcinoma	cemiplimab	I line [125]
Hepatocellular carcinoma	nivolumab/pembrolizumab	II line [126,127]
Cervical cancer	pembrolizumab	II line in tumours with PD-L1 CPS ≥1% [128]
Renal cell carcinoma	nivolumab combined/ pembrolizumab or avelumab combined	I line [129,130,131]
	nivolumab	II line [132]
Classical Hodgkin’s lymphoma	nivolumab	II line [133]
	pembrolizumab	Relapsed after ≥3 lines of therapy [134]
Primary mediastinal B cell lymphoma	pembrolizumab	Relapsed after ≥2 lines of therapy [135]
TNBC	atezolizumab combined	I line if tumours PD-L1^+^ [136]
Endometrial carcinoma	pembrolizumab combined	II line [137]

The approved therapies with anti-PD-1 and anti-PD-L1 are summarized, together with the clinical indications and the references. EGFR: epidermal growth factor receptor; ALK: Anaplastic lymphoma kinase; NSCLS: non small cell lung cancer; MMR: mismatch repair; TNBC: triple negative breast cancer; CPS: combined positive score (number of PD-L1 staining cells (tumour cells, lymphocytes and macrophages) divided by the total number of viable tumour cells, multiplied by 100).

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
