# Peer review of "The Immune Checkpoint PD-1 in Natural Killer Cells: Expression, Function and Targeting in Tumour Immunotherapy"

_cancers, 2020, doi:10.3390/cancers12113285_

Round 1

Reviewer 1 Report

Summary

This is a well-written review article that tackles first and foremost the role of PD-1/PD-L1 axis in NK cell activation in general, and in the setting of solid tumors in particular. In addition, it provides a concise and well-structured summary of the current application, limitations and novel targets of immunotherapy.

Major points

-The authors discuss the expression of PD-1 on benign NK cells, however they do not discuss the expression of PD-L1 by benign NK cells.

Dong et al has recently shown that some tumors can induce PD-L1 on NK cells via AKT signaling, resulting in enhanced NK-cell function and preventing cell exhaustion, and that Anti–PD-L1 mAb directly acts on PD-L1+ NK cells against PD-L1 tumors via a p38 pathway, providing a PD-1–independent mechanism of antitumor efficacy via the activation of PD-L1+ NK cells with anti–PD-L1 mAb (PMID: 31340937).

Could the authors provide a discussion on the expression of PD-L1 on benign NK cells and its' potential therapeutic implications in the setting of solid and hematologic malignancies? Accordingly, we suggest that the authors change the title of section 3 to "Evidence of PD1 and PD-L1 expression on NK cells".

-A brief review of the expression of PD1 & PD-L1 on "malignant" NK cells is suggested as well for completeness sake, as this entails potential therapeutic benefits for NK and NK/T cell leukemias.

The following articles are suggested for citations:

.El Hussein et al have shown expression of PD-L1 in 2/8 cases of EBV-positive aggressive NK cell leukemia (PMID: 32590457)

.Gao et al have shown expression of PD-L1 in 2/4 patients with EBV-negative aggressive NK cell leukemia cases (PMID: 28548121)

.Dufva et al have shown copy gains in PD-L1 & PD-L2 in NK/T cell lymphoma (PMID: 29674644)

Minor points

The manuscript can benefit from light English editing:

-In the abstract, please make the following modifications:

."Thus, the contribution of anti-PD-1/PD-L1 therapy to the recovery of NK cell anti-tumour response.." 

."Here we summarize studies investigating PD-1 expression.."

-In the introduction, please make the following modifications:

."PD1/PDL1 axis"

."that NK cell responses may also.."

-In table 1 legend, please make the following modifications

."The table summarizes the evidence .."

-In section 2, please make the following modifications:

."is yet to be clarified"

."in immune cell inhibition"

-In figure 1 :

.please use "blockage of" instead of "block of"

-in section 3, please make the following modifications:

."On the other hand"

."self-induced"

."murine CMV (MCMV) infection"

."Several evidence demonstrate.."

."PD1/PD-L1 activity"

-in section 4, please make the following modifications:

."Surface receptors"

."Inhibits NK cell activation"

."GR signaling"

."Similarly to what was shown for NK cells.."

."This study on human NK cells also confirmed that.."

."Unlike NK cells"

."PD-1/PD-L1 pathway"

-in section 5.2, please make the following modification:

."Growing evidence"

Author Response

Major points

-The authors discuss the expression of PD-1 on benign NK cells, however they do not discuss the expression of PD-L1 by benign NK cells. 

Dong et al has recently shown that some tumors can induce PD-L1 on NK cells via AKT signaling, resulting in enhanced NK-cell function and preventing cell exhaustion, and that Anti–PD-L1 mAb directly acts on PD-L1+ NK cells against PD-L1 tumors via a p38 pathway, providing a PD-1–independent mechanism of antitumor efficacy via the activation of PD-L1+ NK cells with anti–PD-L1 mAb (PMID: 31340937).

Could the authors provide a discussion on the expression of PD-L1 on benign NK cells and its' potential therapeutic implications in the setting of solid and hematologic malignancies? Accordingly, we suggest that the authors change the title of section 3 to "Evidence of PD1 and PD-L1 expression on NK cells".

We thank the reviewer for this important suggestion. We discussed the expression of PD-L1 by “non malignant” NK cells and its potential therapeutic implications in lines 214-221. We also changed the title of section 3 as suggested (line 120).

-A brief review of the expression of PD1 & PD-L1 on "malignant" NK cells is suggested as well for completeness sake, as this entails potential therapeutic benefits for NK and NK/T cell leukemias. 

The following articles are suggested for citations:

El Hussein et al have shown expression of PD-L1 in 2/8 cases of EBV-positive aggressive NK cell leukemia (PMID: 32590457)

Gao et al have shown expression of PD-L1 in 2/4 patients with EBV-negative aggressive NK cell leukemia cases (PMID: 28548121)

Dufva et al have shown copy gains in PD-L1 & PD-L2 in NK/T cell lymphoma (PMID: 29674644) 

We included a brief review of the expression of PD-L1 on “malignant” NK cells as suggested, citing the recommended articles in lines 210-214.

Minor points

The manuscript can benefit from light English editing: 

-In the abstract, please make the following modifications:

."Thus, the contribution of anti-PD-1/PD-L1 therapy to the recovery of NK cell anti-tumour response.."  this was corrected in line 17

."Here we summarize studies investigating PD-1 expression.." this was corrected in line 18

-In the introduction, please make the following modifications:

."PD1/PDL1 axis" this was corrected in line 50

."that NK cell responses may also.." this was corrected in line 51

-In table 1 legend, please make the following modifications

."The table summarizes the evidence .."  this was corrected in line 236

-In section 2, please make the following modifications:

."is yet to be clarified" this was corrected in line 85

."in immune cell inhibition" this was corrected in line 93

-In figure 1 :

.please use "blockage of" instead of "block of" this was corrected in the figure

-in section 3, please make the following modifications:

."On the other hand" this was corrected in line 124

."self-induced" this was not corrected in line 156 because the induction of self ligands by tumor cells is often referred to as “induced-self” to distinguish it from “missing-self” in NK cell biology.

."murine CMV (MCMV) infection" this was corrected in line 180

."Several evidence demonstrate.." this was corrected in line 200

."PD1/PD-L1 activity" this was corrected in line 235

-in section 4, please make the following modifications:

."Surface receptors" this was corrected in line 248

."Inhibits NK cell activation" this was corrected in line 250

."GR signaling" this was corrected in line 264

."Similarly to what was shown for NK cells.." this was corrected in line 284

."This study on human NK cells also confirmed that.." this was corrected in line 285

."Unlike NK cells" this was corrected in line 290

."PD-1/PD-L1 pathway" this was corrected in line 302

-in section 5.2, please make the following modification:

."Growing evidence" this was corrected in line 375

Reviewer 2 Report

 A good review of synthesis on a rather hot topic. I really enjoyed the in-depth chapter on PD-1 expression on NK and the differences that exist with its expression onT cell  with respect to the stimulus and its kinetics and also the interspecies variations. The role of glucocorticoids in the induction of PD-1 on NK is still poorly understood and has potentially important consequences.

Some suggestions to improve the impact of this review

- long chapter on clinical indications can be reduced. Even if the last part on the role and activity of sPD-1 is a good synthesis of the subject, we are a little off topic with respect to NKs cells.  Are NKs involved in the production of these soluble receptors?

- Apart from PD-1, is there a particular ICP profile for NKs compared to T cells?

- The work of D Olive's group and others showing that NKs in the tumor microenvironment have a dysfunctional phenotype would deserve to complete this review and extend it to a synthesis on the NK phenotype in the tumor microenvironment outside PD-1.

- The possible role of corticosteroids in anti-PD-1/L1 resistance is mentioned. Some articles could be mentioned to complete this paragraph (Ricciuti B J Clin Oncol 2019)

Author Response

A good review of synthesis on a rather hot topic. I really enjoyed the in-depth chapter on PD-1 expression on NK and the differences that exist with its expression on T cell  with respect to the stimulus and its kinetics and also the interspecies variations. The role of glucocorticoids in the induction of PD-1 on NK is still poorly understood and has potentially important consequences.

We thank the reviewer for this very positive comment.

Some suggestions to improve the impact of this review

- long chapter on clinical indications can be reduced. Even if the last part on the role and activity of sPD-1 is a good synthesis of the subject, we are a little off topic with respect to NKs cells.  Are NKs involved in the production of these soluble receptors?

We thank the reviewer for this comment. We removed the description of the clinical indications for anti-PD-(L)1 mAbs from the 5.1 paragraph and summarized it in the new Table 2. Moreover, we modified the 5.2 paragraph and specified the role of NK cells in sPD-1 production.

- Apart from PD-1, is there a particular ICP profile for NKs compared to T cells?

We agree with the reviewer that a comparison of ICP profile (apart from PD-1) between NK and T cells would be interesting. Although it is beyond the scope of the present review, we extended the discussion of ICPs expressed by the two cell types in lines 42-46 adding NKG2A, as it was found co-expressed with PD-1.

- The work of D Olive's group and others showing that NKs in the tumor microenvironment have a dysfunctional phenotype would deserve to complete this review and extend it to a synthesis on the NK phenotype in the tumor microenvironment outside PD-1.

We agree with this important suggestion. We discussed more in general NK cell alterations in the tumor microenvironment in lines 30-37.

- The possible role of corticosteroids in anti-PD-1/L1 resistance is mentioned. Some articles could be mentioned to complete this paragraph (Ricciuti B J Clin Oncol 2019)

We agree with the reviewer that this is an important point. We mentioned the studies by Ricciuti and Arbour, and discussed the possible role for GCs in anti-PD1/PD-L1 resistance in lines 294-300.

Reviewer 3 Report

The authors of this review manuscript provide a comprehensive overview of the studies evaluating the expression of PD-1 in NK cells, its function and immunotherapy strategies. Overall, this work summarises the current evidence in the field but it could be further refined based on some points and comments hereunder:

1. A major question is why the authors focus only in the expression, function and role of PD-1 on NK-cells. Given that:                                                               i) the authors refer to the  targeting of the PD-1/PD-L1 axis which includes also the ligand (e.g. monoclonal antibodies such as atezolizumab against PD-L1)                                                                                              ii) the previously demonstrated important role of PD-L1 expressing NK cells in killing PD-L1-negative tumors cells (Wenjuan Dong et al., Cancer Discovery 2019), thus mediating effective antitumor immune response,                            it would be plausible that the authors include also the expression and role of PD-L1 in NK cells.

2. Table 1: Which are the treatment that the patients have received? Have the patients in these studies been treated with immune checkpoint inhibitors? As a general comment, the authors should refer to potential predictive implications of PD-1 expression NK cells to ICB or other treatment types. It would be therefore  useful to provide such information in the table (+number of patients, % of expression, methods used) and include also some recent articles 

3. Section 5.1: Difficult to follow due to the sequential narrative documentation of the immunotherapeutic indications. I suggest that this can be improved by creating a Table as an alternative way of presentation, since this is not the main focus of this review (not seen here and similarly for sPD-1 section).

4. The abstract needs re-phrasing. It is not clear what is recovery of NK cell responses means and this concept is being introduced in such way for  the first time here. 

5. Introduction: References are lacking regarding the description of the role of NK cells, which should be added

Author Response

The authors of this review manuscript provide a comprehensive overview of the studies evaluating the expression of PD-1 in NK cells, its function and immunotherapy strategies. Overall, this work summarises the current evidence in the field but it could be further refined based on some points and comments hereunder:

  1. A major question is why the authors focus only in the expression, function and role of PD-1 on NK-cells. Given that:
  2. i) the authors refer to the  targeting of the PD-1/PD-L1 axis which includes also the ligand (e.g. monoclonal antibodies such as atezolizumab against PDL1)
  3. ii) the previously demonstrated important role of PD-L1 expressing NK cells in killing PD-L1-negative tumors cells (Wenjuan Dong et al., Cancer Discovery 2019), thus mediating effective antitumor immune response, it would be plausible that the authors include also the expression and role of PD-L1 in NK cells.

We agree with the reviewer that the function and role of PD-L1 on NK cells should also be considered in this review. We discussed the expression of PD-L1 by NK cells and its potential therapeutic implications in lines 210-221, and we changed the title of section 3 accordingly (line 120).

  1. Table 1: Which are the treatment that the patients have received? Have the patients in these studies been treated with immune checkpoint inhibitors? As a general comment, the authors should refer to potential predictive implications of PD-1 expression NK cells to ICB or other treatment types. It would be therefore  useful to provide such information in the table (+number of patients, % of expression, methods used) and include also some recent articles 

We included more details in the Table 1: number of patients, % of PD-1 expression and method of PD-1 detection.

  1. Section 5.1: Difficult to follow due to the sequential narrative documentation of the immunotherapeutic indications. I suggest that this can be improved by creating a Table as an alternative way of presentation, since this is not the main focus of this review (not seen here and similarly for sPD-1 section).

We thank the reviewer for this suggestion. We presented the summary of the therapeutic indications for anti-PD-(L)1 mAbs in the new Table 2.

  1. The abstract needs re-phrasing. It is not clear what is recovery of NK cell responses means and this concept is being introduced in such way for the first time here. 

We rephrased the abstract to make clear what “recovery of NK cell response” means.

  1. Introduction: References are lacking regarding the description of the role of NK cells, which should be added

We added references 1-10 in the introduction.

Reviewer 4 Report

The manuscript by Quatrini et al discusses the potential role of NK cells for efficacy of anti-PD-1/anti-PD-L1 check-point blockade. This is an important area of research, which might explain why check-point inhibition may function also in tumors showing low HLA I expression.

I have some comments below to the various parts of the ms:

1. PD-1 immune checkpoint section

In one sentence the authors write: “Triggering of PD-1 promotes the insurgence of an exhaustion phenotype in both innate and adaptive immune cells…” I wonder which innate immune cells that they refer to - NK cells and ILC or also phagocytes? Please write this in a clearer manner.

In the same section it is said that “PD-1 signaling modulates immune cells metabolism towards a more oxidative environment” Do they mean "towards a more oxidative cell stage"? How does this happen? Please explain, and provide a reference.

2. PD-1 expression on NK cells:

The authors write “High levels of PD-L1 expression have been detected in tumors with low HLA 1 expression, which, in some instances are responsive to PD-1/PD-L1 blockade (ref 28-30)” Since this is a critical point, favoring the hypothesis the PD-1 expression on NK cells may be of clinical relevance, I suggest the authors to present relevant examples from clinical trials where tumors with low/absent HLA I expression have responded to anti-PD-1/anti-PD-L1 therapy.

3. Targeting the PD-1/PD-L1 pathway in tumor immunotherapy

This section thoroughly goes through the clinical usage of anti-PD-1/anti-PD-L1 in various cancers, and the possible prognostic role of soluble PD-1 from a general perspective. Since this review is supposed to discuss the possibility of anti-PD-1/anti-PD-L1 therapy to boost anti-tumor NK cell activities, I suggest rewriting this section to more discuss the clinical use of anti-PD-1/anti-PD-L1 and prognostic role of soluble PD-1 from a NK cell point of view.

Author Response

The manuscript by Quatrini et al discusses the potential role of NK cells for efficacy of anti-PD-1/anti-PD-L1 check-point blockade. This is an important area of research, which might explain why check-point inhibition may function also in tumors showing low HLA I expression.

I have some comments below to the various parts of the ms:

  1. PD-1 immune checkpoint section

In one sentence the authors write: Triggering of PD-1 promotes the insurgence of an exhaustion phenotype in both innate and adaptive immune cells…” I wonder which innate immune cells that they refer to - NK cells and ILC or also phagocytes? Please write this in a clearer manner.

We thank the reviewer for this comment. We specified that we refer to NK and ILCs as the innate immune cells exhausted by PD-1 expression in lines 91-92. 

In the same section it is said that “PD-1 signaling modulates immune cells metabolism towards a more oxidative environment” Do they mean "towards a more oxidative cell stage"? How does this happen? Please explain, and provide a reference.

We thank the reviewer for this comment. We explained how activation of the PD-1 axis affects immune cells metabolism in lines 98-103.

  1. PD-1 expression on NK cells:

The authors write “High levels of PD-L1 expression have been detected in tumors with low HLA 1 expression, which, in some instances are responsive to PD-1/PD-L1 blockade (ref 28-30)” Since this is a critical point, favoring the hypothesis the PD-1 expression on NK cells may be of clinical relevance, I suggest the authors to present relevant examples from clinical trials where tumors with low/absent HLA I expression have responded to anti-PD-1/anti-PD-L1 therapy.

We agree with the reviewer that this is an important point. We reported the example of Hodgkin’s Lymphoma as a tumour with low/negative HLA-I expression (Roemer 2016) and the clinical results of the study by Ansell in lines 146-152.

  1. Targeting the PD-1/PD-L1 pathway in tumor immunotherapy

This section thoroughly goes through the clinical usage of anti-PD-1/anti-PD-L1 in various cancers, and the possible prognostic role of soluble PD-1 from a general perspective. Since this review is supposed to discuss the possibility of anti-PD-1/anti-PD-L1 therapy to boost anti-tumor NK cell activities, I suggest rewriting this section to more discuss the clinical use of anti-PD-1/anti-PD-L1 and prognostic role of soluble PD-1 from a NK cell point of view.

We thank the reviewer for this comment. We re-wrote section 5 to discuss more the clinical use of anti-PD-1/anti-PD-L1 and the role of soluble PD-1 from a NK cell point of view. We also included a new Table (Table 2) to summarize the clinical indications for anti-PD-(L)1 mAbs in many cancer settings.